# Gate-tunable quantum pathways of high harmonic generation in graphene

Soonyoung Cha[1,9], Minjeong Kim[1,2,9], Youngjae Kim[3,9], Shinyoung Choi[1,4], Sejong Kang[5], Hoon Kim[6,7], Sangho Yoon[1,2], Gunho Moon[1,2], Taeho Kim[1,2], Ye Won Lee[1,2], Gil Young Cho [6,7,8], Moon Jeong Park [5], Cheol-Joo Kim [1,4], B. J. Kim[6,7], JaeDong Lee [3] ✉, Moon-Ho Jo [1,2,7] ✉ & Jonghwan Kim [1,2,7] ✉

Under strong laser fields, electrons in solids radiate high-harmonic fields by travelling through quantum pathways in Bloch bands in the sub-laser-cycle timescales. Understanding these pathways in the momentum space through the high-harmonic radiation can enable an all-optical ultrafast probe to observe coherent lightwave-driven processes and measure electronic structures as recently demonstrated for semiconductors. However, such demonstration has been largely limited for semimetals because the absence of the bandgap hinders an experimental characterization of the exact pathways. In this study, by combining electrostatic control of chemical potentials with HHG measurement, we resolve quantum pathways of massless Dirac fermions in graphene under strong laser fields. Electrical modulation of HHG reveals quantum interference between the multi-photon interband excitation channels. As the light-matter interaction deviates beyond the perturbative regime, elliptically polarized laser fields efficiently drive massless Dirac fermions via an intricate coupling between the interband and intraband transitions, which is corroborated by our theoretical calculations. Our findings pave the way for strong-laser-field tomography of Dirac electrons in various quantum semimetals and their ultrafast electronics with a gate control.

Strong laser fields give rise to extreme nonlinear optical phenomena in atoms – high harmonic generation (HHG)[1,2]. In the gas phase, lightwave-driven (or laser-field-driven) electrons radiate intense high-order harmonics through the three-step re-collision processes of tunnel-ionization, free acceleration, and recombination[3,4], which has offered the fundamental basis for attosecond science and technology for the last decades[5]. More recently, since the first discovery of solid-state HHG in a ZnO crystal[6], high harmonic radiation in solids has

emerged as a fascinating new route to probe electronic states in sub-laser-cycle timescale[7–18] and to realize compact, coherent light sources with high photon energy (up to ~40 eV)[8]. In particular, the lightwave-driven quantum pathways of electrons in the momentum space can be visualized by analyzing the spectrum and polarization of high harmonics, which allows to reconstruct band structures[9,10], measure electronic properties such as Berry curvature[11–13] and Chern number[14,15], and observe coherent quasi-particle dynamics in sub-cycle

[1]Center for Van der Waals Quantum Solids, Institute for Basic Science (IBS), Pohang, Republic of Korea. [2]Department of Materials Science and Engineering, Pohang University of Science and Technology, Pohang, Republic of Korea. [3]Department of Physics and Chemistry, Daegu Gyeongbuk Institute of Science and Technology (DGIST), Daegu, Republic of Korea. [4]Department of Chemical Engineering, Pohang University of Science and Technology, Pohang, Republic of Korea. [5]Department of Chemistry, Pohang University of Science and Technology, Pohang, Republic of Korea. [6]Center for Artificial Low Dimensional Electronic Systems, Institute for Basic Science (IBS), Pohang, Republic of Korea. [7]Department of Physics, Pohang University of Science and Technology, Pohang, Republic of Korea. [8]Asia Pacific Center for Theoretical Physics, Pohang, Republic of Korea. [9]These authors contributed equally: Soonyoung Cha, Minjeong Kim, Youngjae Kim. ✉e-mail: jdlee@dgist.ac.kr; mhjo@postech.ac.kr; jonghwankim@postech.ac.kr

timescale[16–18]. However, unlike the gas phase, delocalized electrons in solids can travel complicated multiple quantum pathways via interband and intraband transitions[7,19–21], which sensitively depends on chemical potential and bandgap size under a given laser excitation condition[22]. In semiconductors with a large bandgap, the complication can be effectively addressed by high-order sideband generation[10,16–18] where one laser pulse prepares coherent interband excitation of electron-hole pairs (or excitons) and another laser pulse at terahertz frequencies strongly drives their intraband motion. Unfortunately, it is challenging to avoid the complication for semimetals and narrow bandgap semiconductors with high-order sideband generation because the charge carriers can be readily excited via the interband transition across a small bandgap under strong laser fields even at terahertz frequencies[13,23–25].

Graphene is a two-dimensional semimetal (or a zero-bandgap semiconductor) with linear energy dispersion hosting massless Dirac fermions[26]. Electrons in graphene exhibit strong light-matter interaction due to its unusual band structure. As a linear optical response, the interband and intraband transitions lead to a high absorption coefficient over broad spectral range from terahertz to ultraviolet frequencies[27,28]. The optical resonances from the interband transition enhance nonlinear optical responses[29–32], including third harmonic generation and four-wave mixing. As laser intensity increases over ~1 GW cm$^{-2}$, collective oscillation of massless Dirac fermions via the intraband transition enables remarkably efficient HHG at terahertz frequencies[33,34]. According to a recent experimental study under extreme laser intensity (~ 1 TW cm$^{-2}$)[23], elliptically polarized excitation can further enhance HHG via efficient pair generation mechanism from light-induced transient band structure modification, which is not possible in the standard re-collision model for the gas phases. Theoretical studies suggest other intriguing pathways available in graphene for HHG processes[22,35–42]. On the other hand, the interband and intraband transitions in graphene can be drastically controlled by electrostatically injecting carriers[28,43,44] because the two-dimensional linear band with the high Fermi velocity (~10$^6$ m s$^{-1}$) allows large modification of the chemical potential by small amount of carrier injection. Therefore, graphene provides an ideal platform to resolve the interplay between the interband and intraband transitions in quantum pathways of carriers under strong laser fields.

In this work, we investigate widely tunable HHG in graphene with electrostatic control of the chemical potential to resolve quantum pathways of massless Dirac fermions under strong laser fields. Under linearly-polarized excitation with laser intensity of ~1 GW cm$^{-2}$, harmonic intensity exhibits non-monotonic dependence on the chemical potential, which arises from the destructively interfering excitation channels via the multi-photon interband transitions. Under elliptically polarized excitation, as the light-matter interaction deviates beyond the perturbative regime, charge carriers excited by the multi-photon transitions efficiently radiate the high harmonics with the drastically rotated polarization axis, which is completely different from the previous study[23]. Interestingly, the helical response is substantially suppressed by blocking the interband transitions. Our full quantum mechanical theory identifies a characteristic mechanism for massless Dirac fermions where the intricate coupling between interband and intraband transitions efficiently drives charge carrier motion under elliptically polarized excitation. Our study paves the way for visualizing sub-cycle dynamics of massless charge carriers in quantum materials including graphene, topological insulators and Weyl semimetals, and realizing their ultrafast electronics and attosecond photonics with an electrostatic control.

## Results and discussion
### HHG measurement with a gate control of chemical potentials
Figure 1a–c schematically illustrate the electronic processes in graphene under strong laser fields. In the real space (Fig. 1a), laser field

($\mathbf{E}(t)$) centered at the frequency of $\omega$ excites and transports charge carriers to generate an ultrafast anharmonic current ($\mathbf{J}(t)$) which radiates high harmonics ($\mathbf{I}^{(n\omega)}$)[7,45], where $n$ is a positive integer to represent harmonic orders. In the momentum space (Fig. 1b, c), $\mathbf{J}(t)$ can be generated from charge carriers following the multiple quantum pathways which are composed of the interband and intraband transitions. The control of the chemical potential ($\mu$) leads to a profound impact on the individual quantum pathway. For the charge-neutral case ($\mu = 0$) in Fig. 1b, where $\mu$ is at the Dirac point or the charge-neutral point, charge carriers in the pathways can be excited and recombined via linear and nonlinear optical transitions between the valence band and the conduction band (black solid arrow). The photo-excited carriers can also be driven via the intraband transition simultaneously (red and blue solid arrows). For the hole-doped case ($\mu < 0$) in Fig. 1c, the interband transition (black dashed arrow) is forbidden for the empty states between the Dirac point and $\mu$, which does not create photo-excited carriers[28,43,44]. Subsequently, the laser field cannot drive charge carriers via the connected intraband transitions (red and blue dashed arrows). For the electron-doped case ($\mu > 0$), the same pathways are not available either since the interband transition is blocked for the filled states between the Dirac point and $\mu$. On the other hand, the laser field can drive electrostatically injected carriers to generate $\mathbf{J}(t)$ solely via intraband transition without interband transition[34,46]. Therefore, control of $\mu$ allows to understand distinct contributions of different quantum pathways to the HHG process in graphene.

The experimental configurations are schematically described in Fig. 1a. Graphene grown by chemical vapor deposition method (CVD) is transferred on a sapphire substrate. The device structure of a field-effect transistor is fabricated on graphene to control and characterize $\mu$. The source electrode (S) and the drain electrode (D) measure static current along the graphene channel under the bias voltage ($V_b$). Ion-gel covers the gate electrode (G) and graphene to electrostatically inject electrons or holes into graphene by applying the gate voltages ($V_G$). Intense mid-infrared femtosecond laser pulses ($\mathbf{E}(t)$) are irradiated on graphene with elliptical polarization. The polarization major axis (the x-direction) and the minor axis (the y-direction) are assigned parallel and perpendicular, respectively, to the direction along the source and the drain electrodes of the devices. The ratio of two orthogonal electric fields ($E_y/E_x$), defined as ellipticity ($\varepsilon_{exc}$), is controlled by a liquid crystal variable retarder. Combination of the polarization analysis optics, the spectrometer, and EMCCD measures and analyzes detailed characteristics of transmitted harmonics ($\mathbf{I}^{(n\omega)}$) including their spectra, intensity, and polarization. More detailed configuration is provided in Supplementary Note.

High harmonic spectrum (Fig. 1d) shows the fifth and seventh harmonics at 1.38 eV and 1.93 eV, respectively, under laser excitation at 0.28 eV (see Supplementary Fig. 1 for subtraction of background luminescence). Laser peak intensity ($I_{exc}$) dependence indicates that the non-perturbative response emerges in generation of the harmonic signals under strong laser fields. Under $\varepsilon_{exc} = 0$ (i.e., linearly-polarized along the x-direction), intensity of the fifth harmonics (black circles in Fig. 1e) scales initially to the fifth power of $I_{exc}$ (blue line), which is expected from the perturbative response[47], but saturating roughly to the second power of $I_{exc}$ (red line) as $I_{exc}$ increases from 1.8 to 55 GW cm$^{-2}$. In addition, elliptically polarized excitation with $\varepsilon_{exc} = 0.3$ reveals unusual polarization profile of harmonics. Figure 1g displays a polar plot of the fifth harmonic intensity which is measured after a linear polarizer as a function of the polarizer angle ($\varphi$). The orientation of the sample and the major axis of the laser polarization are fixed along the x-direction ($\varphi = 0°$) as described in Fig. 1a, f, respectively. The polarization axis of the harmonic signals in Fig. 1g is rotated counterclockwise by 24.9° for 2.4 GW cm$^{-2}$ in comparison to the laser polarization axis (Fig. 1f). Higher $I_{exc}$ rotates the polarization axis of the harmonic signals even further, which cannot be expected from the perturbative response[47] (see Supplementary

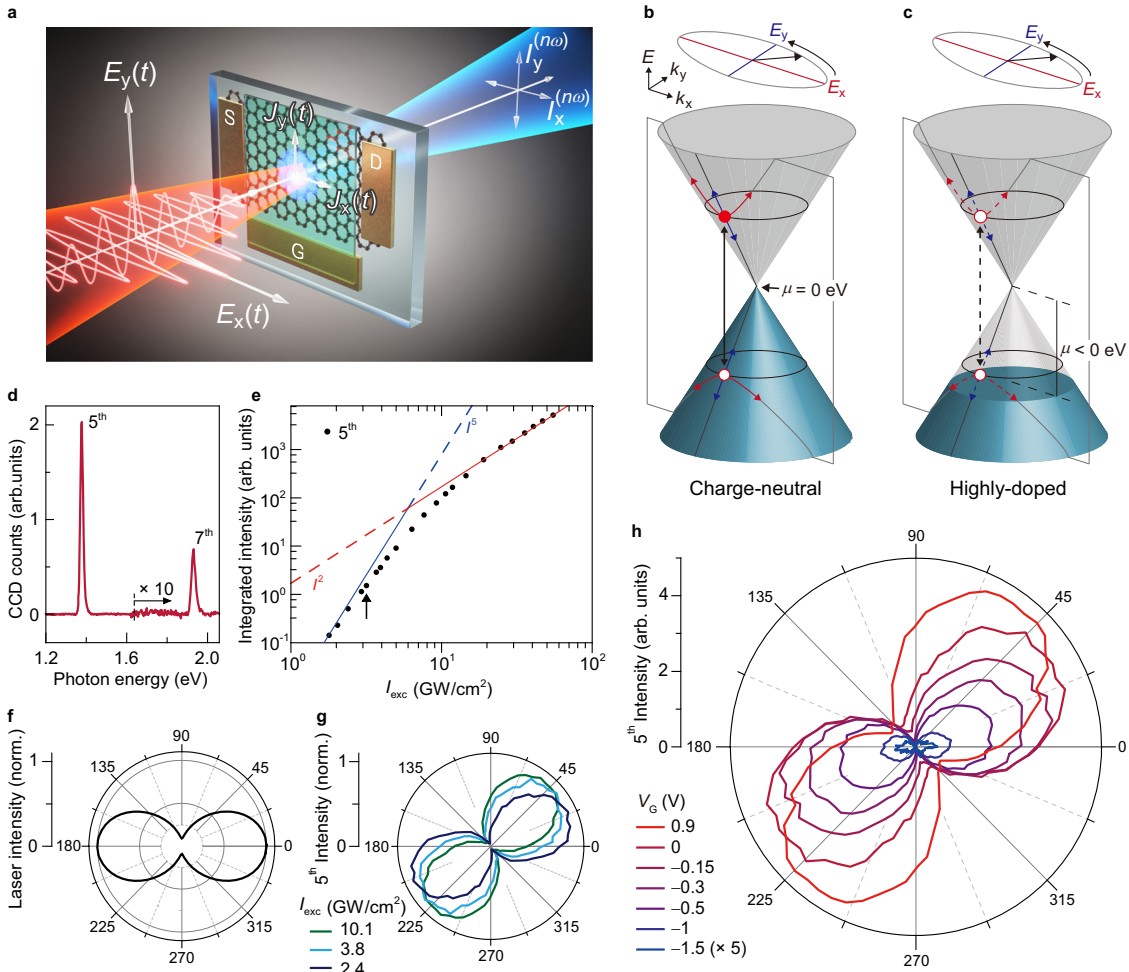

**Fig. 1 | Gate-tunable HHG in graphene. a** Schematic of HHG measurement. An intense mid-infrared femtosecond laser pulse $\mathbf{E}(t)$ generates an anharmonic ultrafast current $\mathbf{J}(t)$ in the graphene device with ion-gel gating, radiating high harmonics $\mathbf{I}^{(n\omega)}$. **b, c** Schematic of chemical potential ($\mu$) dependent HHG process in momentum space. For the charge-neutral case (**b**), $\mathbf{J}(t)$ is generated simultaneously by the interband transition (black solid arrow), intraband transition along the $x$-direction (red solid arrow), and intraband transition along the $y$-direction (blue solid arrow). For the highly doped case, the interband transition (black dashed arrow) and connected intraband transitions (red and blue dashed lines) are blocked due to Pauli blocking. **d** HHG spectrum. The fifth and seventh harmonics are observed at 1.38 and 1.93 eV, respectively, under excitation photon energy of

0.28 eV. **e** Intensity of the fifth harmonics as a function of the laser peak intensity $I_{exc}$. The harmonic intensity (black circles) initially scales with the fifth power of $I_{exc}$ (blue line), but it gradually saturates to the second power (red line) of $I_{exc}$ as increasing $I_{exc}$. Black arrow indicates the intensity $I_{exc}$ of 3.1 GW/cm². **f** Laser polarization profile with ellipticity $\varepsilon_{exc}$ of 0.3. **g** Normalized polarization profile of the fifth harmonics under elliptically polarized excitation (**f**) with different $I_{exc}$. **h** Polarization profile of the fifth harmonics with gate voltage control $V_G$ under $\varepsilon_{exc} = 0.3$. For clarity, polarization plot for $V_G = -1.5$ V is multiplied by a factor of 5. As $V_G$ decreases from 0.9 V to −1.5, the harmonic intensity is reduced by ~50 times while the polarization axis is rotated by 52.7°.

Note for $I_{exc}$ - dependence of the fifth harmonic expected from perturbative nonlinear optics). The polar plots of harmonics intensity and laser intensity are normalized to clearly visualize the rotation angles. Despite low $I_{exc}$ of a few GW cm⁻², we observe similar behavior of the fifth harmonics to the experimental results in the previous study[23] with high $I_{exc}$ of ~1 TW cm⁻². Surprisingly, however, we observe that application of $V_G$ drastically modifies the fifth harmonics (Fig. 1h). As $V_G$ is varied from 0.9 V to −1.5 V for the same $I_{exc}$ of 3.1 GW cm⁻² (marked with black arrow in Fig. 1e), harmonic intensity is reduced by ~50 times while polarization axis is rotated from 52.7° to nearly 0°. The widely tunable harmonic signals with $V_G$ suggest that charge carriers indeed undergo different pathways of harmonic generation depending on $\mu$.

In order to calibrate our devices from $V_G$ to $\mu$, we measure both electrical transport and infrared transmittance spectra of graphene. Blue line in Fig. 2a indicates that the maximum resistance is achieved at $V_G = 0.95$ V, identifying as the charge-neutral point ($\mu = 0$ eV). Resistance sharply drops for $V_G < 0.95$ V (or $V_G > 0.95$ V) due to carrier

injection of holes (or electrons). $\mu$ away from the Dirac point can be identified by infrared transmission spectroscopy which probes modulation of interband optical transition[28,43,44]. In the linear optical process, interband transition is forbidden by Pauli blocking for photon energy below $2|\mu|$. Real part of optical conductivity shows a step-like absorption edge at photon energy of $2|\mu|$, and Fig. 2b demonstrates such step-like profiles with broadening from scattering processes. The absorption edge gradually moves to higher energy as $\mu$ is lowered from the Dirac point. Fitting to the model based on Kubo formula provides $\mu$ which are summarized with black empty squares in Fig. 2a. $\mu$ measured from the electrical and optical measurements shows good agreement for ion-gel gating of graphene with a capacitance 4.8 µF cm⁻² (see Supplementary Note for more details on the experimental determination of $\mu$). In this study, we focus on the hole-doped region ($\mu < 0$) which provides wider electro-chemical window for our graphene devices without chemical degradation by ion-gel. Nevertheless, the physical mechanism should be identical for the electron-doped region ($\mu > 0$) as discussed above.

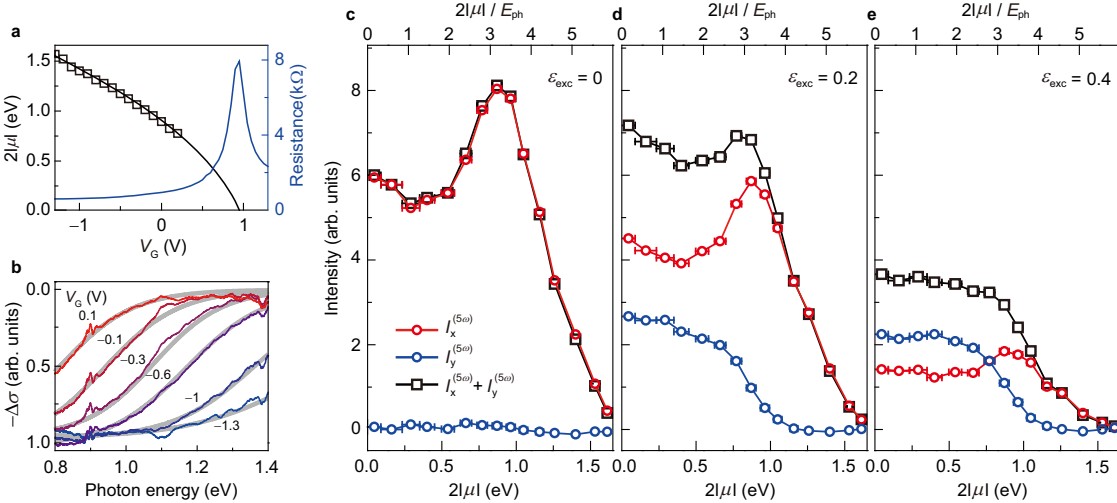

**Fig. 2 | Intensity of the fifth harmonics as a function of the chemical potential (μ). a** Electrical transport measurement of the graphene channel. Resistance (blue solid line) shows the maximum at the gate voltage $V_G$ of 0.95 V, identifying the charge-neutral case ($\mu = 0$ eV). **b** Infrared transmission measurement of the graphene device. As $V_G$ decreases from 0.1 V to −1.3 V, a step-like absorption edge moves to higher energy. Fitting to the model based on Kubo formula (gray solid lines) determines $\mu$ which is summarized in **a** with black squares. **c**–**e** Harmonic intensity as a function of $2|\mu|$ (or $2|\mu|/E_{ph}$ where $E_{ph}$ is laser photon energy). $I_x^{(5\omega)}$ (red circles) and $I_y^{(5\omega)}$ (blue circles) represent harmonic intensity along the $x$-direction and $y$-direction, respectively. $I_x^{(5\omega)} + I_y^{(5\omega)}$ (black squares) represents total intensity of the harmonics. Under linearly-polarized excitation (**c**), $I_x^{(5\omega)}$ shows a resonance-like profile with the maximum intensity as $2|\mu| = 0.87$ eV while $I_y^{(5\omega)}$ is absent beyond the noise level. Under elliptically polarized excitation with ellipticity $\varepsilon_{exc} = 0.2$ (**d**) and $\varepsilon_{exc} = 0.4$ (**e**), the fifth harmonics shows $I_x^{(5\omega)}$ and $I_y^{(5\omega)}$ with a resonance-like profile and a step-like profile, respectively.

## Interband transition channels of HHG

As the simplest case, we examine harmonic generation in graphene under linearly-polarized laser excitation ($\varepsilon_{exc} = 0$). $I_{exc}$ is kept at 3.1 GW cm$^{-2}$ to avoid any possible laser-induced damage during a long-term exposure for accurate optical measurement (see Supplementary Fig. 2 for the laser-induced damage on graphene). In Fig. 2c, the fifth harmonics are observed for the polarization along the x-direction (red circles labeled with $I_x^{(5\omega)}$) with the non-monotonic dependence on the chemical potential. For the y-direction, no signal is observed beyond the noise level (blue circles labeled with $I_y^{(5\omega)}$). We find that the fifth harmonics are linearly-polarized along the parallel direction of the laser polarization regardless of the sample orientation (see Supplementary Fig. 3 for the harmonic signals from different sample orientations). As the chemical potential is lowered, $I_x^{(5\omega)}$ shows a resonance-like profile with the maximum intensity at $2|\mu| = 0.87$ eV. Dramatically small signal for $2|\mu| > 1.5$ eV suggests that the intraband transition of electrostatically injected carriers alone provide minor contribution to HHG in highly doped graphene under $I_{exc}$ of 3.1 GW cm$^{-2}$. Instead, the interband carrier excitation plays an important role in the pathway to generate the fifth harmonics.

Enhancement of $I_x^{(5\omega)}$ at the specific $\mu$ indicates that the interband excitation of charge carriers is dominated by the multi-photon transition. The multi-photon transition rate is determined by the multiple resonant channels with the energy separation of $mE_{ph}$, where $m$ and $E_{ph}$ denote the number of photons involved in the transition and the individual photon energy of laser, respectively[47]. According to the recent studies on the perturbative nonlinear optical process in graphene[29,40,48], these resonant channels can destructively interfere each other while their magnitudes of transition rate are in the similar order of magnitude. As the resonant channels are sequentially switched off by Pauli blocking, the harmonic signal can increase as the destructive interference is partly eliminated. The harmonic signal eventually disappears as all the relevant resonant channels are switched off. Therefore, enhancement of $I_x^{(5\omega)}$ in Fig. 2c can be explained by partial elimination of destructive interference among available interband transition channels via multi-photon transition[40].

Our interpretation can be further supported by measurement with the different laser excitation energies. For the laser excitation energy at 0.31 eV and 0.35 eV, $I_x^{(5\omega)}$ shows similar resonance-like profiles, but the entire profiles are precisely shifted with the laser excitation energies (see Supplementary Fig. 4 for the excitation energy dependence). In principle, harmonics enhanced by multi-photon transition exhibits series of multiple sharp resonance-like profiles at $2|\mu|/E_{ph} = n$, which is expected by theoretical calculation in the perturbative regime[40]. However, we observe a broad resonance-like profile (Fig. 2c) with the maximum intensity at $2|\mu|/E_{ph}$ slightly above 3. This can be due to ultrafast electronic scattering process which drastically broaden and merge series of sharp resonance-like profiles together. (see Supplementary Fig. 5 for the variation of $2|\mu|$ for the maximum $I_x^{(5\omega)}$ over devices and Supplementary Note for the theoretical calculation of $I_x^{(5\omega)}$ as a function of $2|\mu|$ and discussion on the detailed characteristics).

## HHG under elliptically polarized excitation

Harmonic generation in graphene is further examined under elliptically polarized laser excitation. Elliptically polarized excitation allows to probe the interplay between microscopic processes in the generation of harmonics. According to the standard re-collision model for atomic gases[3], rotation of electric field steers tunnel-ionized electrons away from parent ions during the acceleration process, which decreases probability of the recombination process. Harmonic generation efficiency consequently drops with laser ellipticity[49]. For solids with crystalline symmetry, elliptically polarized excitation can enhance HHG efficiency as recently demonstrated for MgO, Si, Bi$_2$Se$_3$, and graphene[19,20,23,50]. Theoretical studies suggest several possible models where the enhancement is due to the increased recombination probability at the neighboring atomic sites[20] or due to the tightly coupled dynamics of the intraband and interband transitions[22,51]. Alternatively, strong laser fields can modify a band structure of solids from a semiconducting structure to a semimetallic structure, which facilitates ultrafast current generation under elliptically polarized excitation[23,52].

Figure 2d and e show the fifth harmonic signals in graphene with laser ellipticity $\varepsilon_{exc} = 0.2$ and 0.4, respectively. While $\varepsilon_{exc}$ is modified,

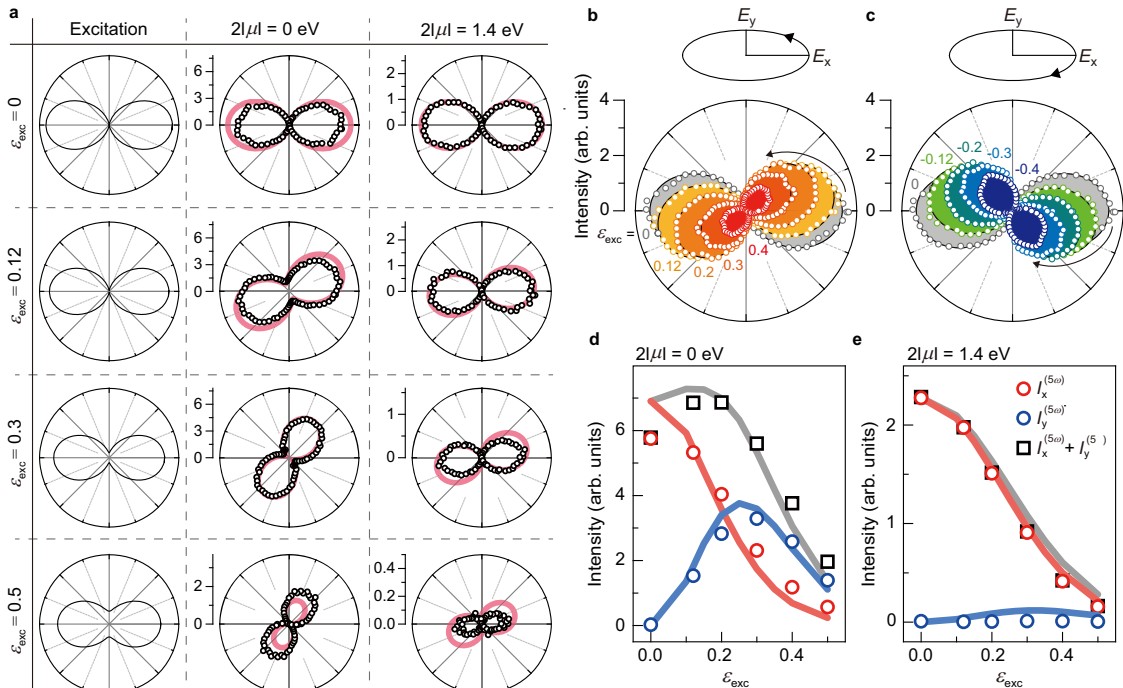

**Fig. 3 | Polarization of the fifth harmonics for the charge-neutral case and highly doped case. a** Polarization profiles of the fifth harmonics. Black circles show the fifth harmonic intensity measured after a linear polarizer as a function of polarizer angle $\varphi$. Polarization profile of laser excitation with ellipticity $\varepsilon_{exc} = 0$, 0.12, 0.3, and 0.5 (left column labeled with Excitation) is plotted as a guidance. For the charge-neutral case (middle column labeled with $2|\mu| = 0$ eV), the harmonic polarization sensitively rotates with $\varepsilon_{exc}$. For the highly doped case (right column labeled with $2|\mu| = 1.4$ eV), the harmonic polarization nearly does not rotate up to $\varepsilon_{exc} = 0.5$. Pink solid lines show the theoretically calculated polarization, which

exhibits good agreement with experimental results. **b, c** Direction of harmonic polarization rotation under elliptically polarized excitation. Under laser excitation with left-handed helicity in (b) (or right-handed helicity in (c)), the polarization axis rotates in counterclockwise direction (or clockwise direction) with $\varepsilon_{exc}$. **d, e** Harmonic intensity as a function of $\varepsilon_{exc}$. Both $I_x^{(5\omega)}$ (red circles) and $I_y^{(5\omega)}$ (blue circles) are observed for the charge-neutral case whereas $I_y^{(5\omega)}$ is nearly absent for the highly doped case. Theoretical calculation results (solid lines) show a good agreement with experimental results.

the pulse energy is set identically to the case of linearly-polarized excitation. The harmonic signals are observed for both polarization directions along the x-direction (red circles) and the y-direction (blue circles). For the charge-neutral case ($2|\mu| = 0$ eV), total intensity of the harmonic signal (black squares) is noticeably increased under the laser excitation of $\varepsilon_{exc} = 0.2$ in comparison to $\varepsilon_{exc} = 0$. This enhancement can be attributed to efficient generation of harmonic signals for $I_y^{(5\omega)}$. Despite small laser fields along the y-direction, elliptically polarized excitation generates surprisingly strong $I_y^{(5\omega)}$. For example, $I_y^{(5\omega)}$ reaches nearly 60 percent of $I_x^{(5\omega)}$ despite 25 times smaller laser intensity along the minor axis than the major axis. Remarkably, $I_y^{(5\omega)}$ is even greater than $I_x^{(5\omega)}$ for laser excitation of $\varepsilon_{exc} = 0.4$. We do not observe any noticeably different behavior for other sample orientations (see supplementary Fig. 6 for the sample orientation dependence of harmonic signals for elliptically polarized excitation). Efficient generation of $I_y^{(5\omega)}$ under elliptically polarized excitation suggests an intriguing pathway in graphene where laser fields $E_x$ and $E_y$ are strongly coupled for the generation of harmonics.

The chemical potential dependence shows that this pathway can be switched off with the $\mu$ control. $I_y^{(5\omega)}$ disappears a step-like profile where intensity decreases roughly across $2|\mu| = 0.87$ eV for both $\varepsilon_{exc} = 0.2$ and 0.4. On the other hand, $I_x^{(5\omega)}$ shows a resonance-like profile with maximum at $2|\mu| = 0.87$ eV as in the case of linearly-polarized excitation (Fig. 2c). Under laser excitation energies at 0.31 eV and 0.35 eV, we observe similar profiles for $I_x^{(5\omega)}$ and $I_y^{(5\omega)}$, but the entire profiles are precisely shifted with the laser excitation energies (see Supplementary Fig. 7 for the excitation energy dependent HHG given by elliptically polarized light). This means that the pathway for the efficient generation of $I_y^{(5\omega)}$ also requires the multi-photon interband excitation of charge carriers. However, the

two contrasting dependences on chemical potential imply that the interplays of microscopic electronic processes are markedly different in the pathways generating $I_x^{(5\omega)}$ and $I_y^{(5\omega)}$.

The pathways of charge carrier in real space can be probed by measuring polarization profile of harmonic signals. Ultrafast current ($\mathbf{J}^{(5\omega)}$) induced by laser field generates harmonic radiation field ($\mathbf{E}^{(5\omega)}$) satisfying the following equation:

$$I_{\hat{\varphi}}^{(5\omega)} \sim |5\omega J_{\hat{\varphi}}^{(5\omega)}|^2 \tag{1}$$

$J_{\hat{\varphi}}^{(5\omega)}$ and $I_{\hat{\varphi}}^{(5\omega)}$ are ultrafast current and harmonic radiation intensity along direction $\hat{\varphi}$ at frequency of $5\omega$, respectively. In our experiment, $I_{\hat{\varphi}}^{(5\omega)}$ can be measured by recording harmonic intensity after a linear polarizer as a function of polarizer angle ($\varphi$). Black empty circles in polar plots of Fig. 3a show polarization of fifth harmonics for the charge-neutral case (column labeled with $2|\mu| = 0$ eV) and the highly doped case (column labeled with $2|\mu| = 1.4$ eV) as the two representative cases with control of $\varepsilon_{exc}$ (see Supplementary Fig. 8 for more measurement at different chemical potentials). The first column (labeled with Excitation) is laser intensity profile which is shown as a guidance to describe the direction of the major axis (the x-direction, $\varphi = 0°$), minor axis (the y-direction, $\varphi = 90°$), and ellipticity of laser excitation. Laser excitation helicity is set as counterclockwise (left-handed) rotation with respect to laser propagation direction. Under linearly-polarized excitation ($\varepsilon_{exc} = 0$), the harmonic signals for both cases ($2|\mu| = 0$ eV and $2|\mu| = 1.4$ eV) exhibit linear polarization along the x-direction, meaning that laser excitation induces $\mathbf{J}^{(5\omega)}$ in graphene along the parallel direction of the laser polarization.

As laser ellipticity $\varepsilon_{exc}$ increases, harmonic signals show completely different behaviors. For the charge-neutral case, harmonic

polarization axis sensitively rotates with ellipticity. Under $\varepsilon_{exc}$ = 0.5, polarization axis is rotated by 58.7° with respect to the laser polarization axis. The rotation direction of polarization axis is determined by laser excitation helicity. Under laser excitation with the left-handed helicity, polarization axis rotates in the counterclockwise direction, which is also shown in Fig. 3b with additional data ($\varepsilon_{exc}$= 0.2 and 0.4). Under right-handed helicity excitation, the polarization axis rotates by exactly the same angles but in clockwise direction (Fig. 3c). This indicates that elliptical laser excitation can rotate the orientation of $\mathbf{J}^{(5\omega)}$ following the laser field helicity. However, polarization axis nearly does not rotate up to $\varepsilon_{exc}$ = 0.5 for the highly doped case (Fig. 3a). The two contrasting behaviors upon $\varepsilon_{exc}$ are summarized in Fig. 3d, e where red circles, blue circles, and black squares represent $I_x^{(5\omega)}$, $I_y^{(5\omega)}$, and $I_x^{(5\omega)} + I_y^{(5\omega)}$, respectively. For $2|\mu| = 0$ eV (Fig. 3d), elliptically polarized excitation efficiently generates $I_y^{(5\omega)}$ in addition to $I_x^{(5\omega)}$. For example, substantially smaller laser field along the y-direction efficiently generates $I_y^{(5\omega)}$ at $\varepsilon_{exc}$ = 0.12, which leads to harmonic signals with remarkably large polarization rotation and ellipticity. On the other hand, $I_y^{(5\omega)}$ is nearly absent for $2|\mu| = 1.4$ eV (Fig. 3e) for all ellipticities, which orients the harmonic signals nearly along the x-direction. Considering the fact that multi-photon interband transition are suppressed upto the five photon transition (1.38 eV) for $2|\mu| = 1.4$ eV due to Pauli blocking, charge carrier excitation via the interband transition also plays an important role for charge carriers to generate $J_y^{(5\omega)}$ efficiently.

In the previous study on graphene[23], the efficient generation of $I_y^{(5\omega)}$ is attributed to transient band structure modification under intense laser excitation (~1 TW cm$^{-2}$). According to the suggested model for a semiconductor[52,53], strong laser field closes a bandgap by lowering a conduction band and raising a valence band, which increases overlap between the two bands. Pairs of electrons and holes in this semimetallized structure can be readily generated without the interband transition. This pair generation mechanism can be particularly facilitated in graphene because the conduction band and valence band overlaps at Dirac point without a bandgap. Their following analysis on charge carrier trajectory shows that elliptically polarized excitation with finite ellipticity can exhibit larger recombination probability of charge carriers than linearly-polarized excitation. However, in our experiment with orders of magnitude lower laser intensity (~1 GW cm$^{-2}$), systematic chemical potential dependence of harmonic signals (Fig. 2c–e) suggests that charge carriers are dominantly excited via the multi-photon transitions instead of the pair generation in the transiently semimetallized band structure. Therefore, we conclude that graphene hosts an additional efficient pathway to generate high harmonics under elliptically polarized excitation in the multi-photon excitation regime.

## Theoretical analysis of HHG in graphene

In order to understand the underlying microscopic mechanism, we solve Liouville-von Neumann equation (see Supplementary Note on Liouville-von Neumann equation) to obtain photo-excited charge carrier population for the charge neutral case where graphene is excited by linearly-polarized laser field along x-direction ($E_x$). Population relaxation rate is assumed negligibly small to visualize photo-excited carrier population under strong laser fields. Figure 4a shows the conduction band population ($\rho_{CB}$) immediately after laser excitation with intensity of 3.1 GW cm$^{-2}$. In the momentum space, population is mostly concentrated around the Dirac points at the K and K′ points where broad low-energy interband transition is available from terahertz to ultraviolet frequencies along Dirac cones. The detailed profile of carrier population around the K point (region marked with a white square) is shown in Fig. 4b. Dashed lines are constant energy contours on a Dirac cone, which describes the conduction band states vertically separated by energy $mE_{ph}$ from the valence band. Carriers are mostly distributed from the

Dirac point to the states on the contour of $5E_{ph}$ along $k_y$-axis while carriers are absent along the $k_x$-axis. For single-photon transition, the pronounced nodal line can be explained by optical selection rule of massless Dirac fermion. Direction of transition dipole moment circulates around the Dirac point, which prohibits interband transition for states with $k_y = 0$ under linearly-polarized laser field along the x-direction ($E_x$) (Fig. 4a). Such characteristic excitation profile from the optical selection rule is also observed in ARPES measurements[54] and ultrafast pump-probe studies[55]. Our numerical calculation in Fig. 4a and b indicates that the multi-photon transition as well as the single-photon transition is also blocked for the states on the $k_x$-axis.

The population relaxation processes quickly relax the anisotropic population (Fig. 4a, b) to a thermalized isotropic population within ~30 fs[54–56], which typically generates the same photocurrent regardless of direction of an external electric field[57]. However, the anisotropic excitation profile plays an important role in high harmonic generation because charge carriers under strong laser fields radiate harmonic fields by interband and intraband transitions in sub-laser-cycle timescale (<15 fs in our experiment)[45]. Figure 4d, e schematically describes the microscopic channels for carrier dynamics at the states on the $k_y$-axis (i.e. $k_x = 0$, $k_y \neq 0$) as the representative cases. Under linearly-polarized excitation along x-direction (Fig. 4d), the interband transition (black arrows labeled with $M_x^{inter}$) creates and recombines photo-excited electrons and holes (red filled and empty circles, respectively). Simultaneously, laser field $E_x$ drives photo-excited carriers via the intraband transition along the $k_x$-direction (orange arrows labeled with $M_x^{intra}$). Under elliptically polarized excitation (Fig. 4e), laser field $E_y$ also drives carriers via the intraband transition along the $k_y$-direction with a phase delay of $\pi/2$ (blue arrows labeled with $M_y^{intra}$). The interband transition from laser field $E_y$ ($M_y^{inter}$) is not allowed for the states as shown in Fig. 4b. The intricate coupling of $M_x^{inter}$ and $M_y^{intra}$ provides an additional pathway to generate ultrafast current that is not available under linearly-polarized excitation. In particular, the population profile from $M_x^{inter}$ (Fig. 4b) can exhibit highly anisotropic response in terms of ultrafast current generation along the two orthogonal directions via $M_x^{intra}$ and $M_y^{intra}$.

The harmonic signals from graphene can be theoretically calculated by solving density matrices with quantum master equation in the Houston basis (see Supplementary Note for more details on theoretical calculation). Under the same laser excitation condition in our experiment, the theoretical calculation (pink solid lines in Fig. 3a, solid lines in Fig. 3d, e) reproduces our experimental results with an excellent agreement. Detailed chemical potential dependence of $I_x^{(5\omega)}$ and $I_y^{(5\omega)}$ also shows the characteristic features from charge carrier excitation via the multi-photon transition, which is consistent with our experimental observation in Fig. 2c–e (see Supplementary Note for calculation of chemical potential dependent HHG).

In order to resolve the individual contributions from different pathways under elliptically polarized excitation, we employ the uniform velocity gauge where Bloch functions and gauges are fully decomposed. In the velocity gauge, light-electron interaction can be written in Hamiltonian, $p \cdot \mathbf{A}(\tau)/c$. $\mathbf{A}(\tau)$ and $c$ are the vector potential of laser field and the speed of light, respectively, and $p$ is the optical matrix element of Dirac electrons in graphene. This representation allows to distinguish charge carrier dynamics naturally via the interband and intraband transitions induced by the time-dependent vector potential of a single external laser pulse. $p \cdot \mathbf{A}(\tau)/c$ can be decomposed into the arbitrarily model gauge based on the interband and intraband transitions under laser fields along the x- and y-directions (i.e., $p_x^{inter}A_{inter}(\tau)/c$, $p_y^{inter}A_{inter}(\tau)/c$, $p_x^{intra}A_{intra}(\tau)/c$, and $p_y^{intra}A_{intra}(\tau)/c$) so that the interaction part of the Hamiltonian can be rewritten by the microscopic channels of $M_x^{inter}$, $M_y^{inter}$, $M_x^{intra}$, and $M_y^{intra}$. One can examine harmonic signals generated from individual decomposed

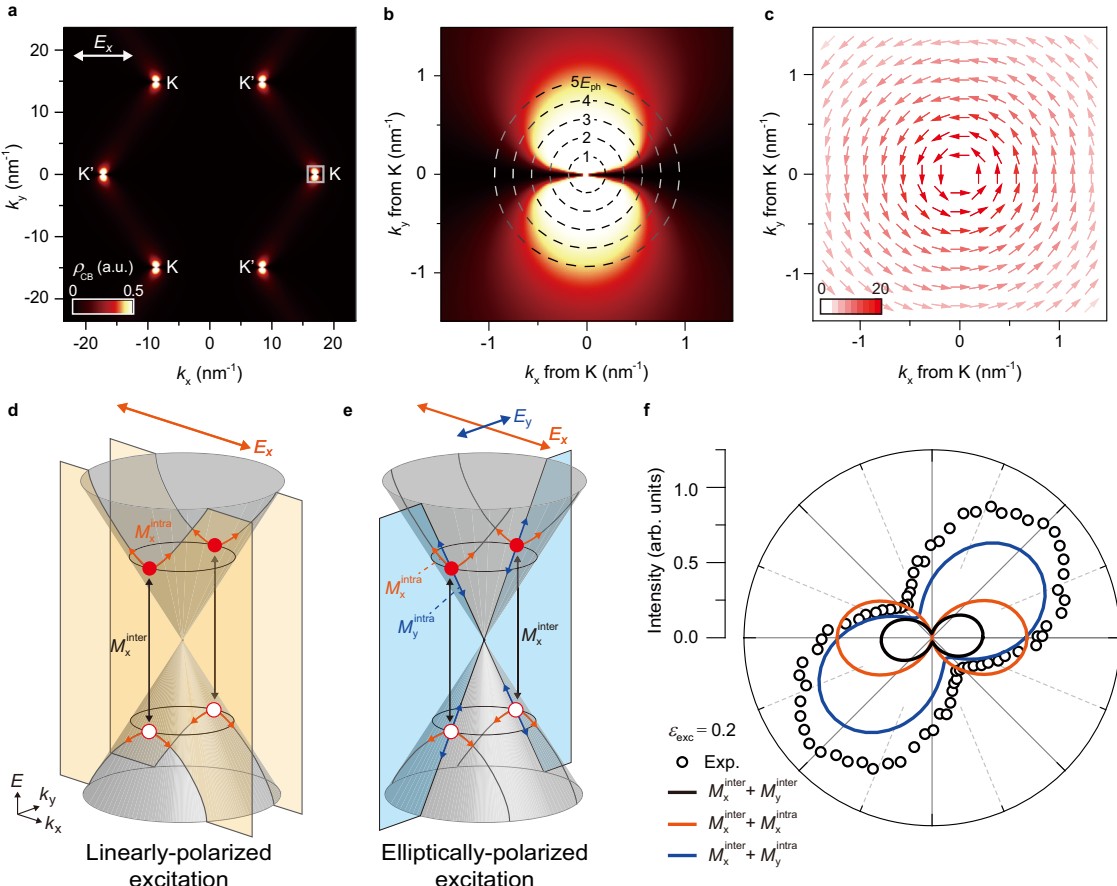

**Fig. 4 | Theoretical calculation of HHG from massless Dirac fermions in graphene. a** Photo-excited charge carrier population in the conduction band $\rho_{CB}$ under linearly-polarized laser field along the x-direction ($E_x$). **b** Detailed profile of carrier population around K point (region marked with a white square in **a**). Dashed lines are constant energy contours on a Dirac cone, which describes conduction band states vertically separated by energy $mE_{ph}$. **c** Transition dipole moment profile. Color scale and arrow show the magnitude and direction of transition dipole moment. **d, e** Microscopic channels for carrier dynamics at the states on the $k_y$-axis. Under linearly-polarized excitation along the x-direction (**d**), the interband transition (black arrows labeled with $M_x^{inter}$) creates and recombine photo-excited electrons and holes (red filled and empty circles, respectively). Simultaneously, the intraband transition drives carriers along the $k_x$-direction (orange arrows labeled with $M_x^{intra}$). Under elliptically polarized excitation (**e**), the intraband transition also drives carriers along the $k_y$-direction (blue arrows labeled with $M_y^{intra}$). **f** Polarization profile of the fifth harmonics under laser ellipticity $\varepsilon_{exc} = 0.2$ from selected pathways. Black, orange, and blue solid lines show the polarization profile from quantum pathways via $M_x^{inter} + M_y^{inter}$, $M_x^{inter} + M_x^{intra}$, and $M_x^{inter} + M_y^{intra}$, respectively. Black empty circles show experimental results for the same laser excitation condition.

pathways selectively by constructing a desired Hamiltonian from linear combination of microscopic channels.

Solid lines in Fig. 4f show polar plots of fifth harmonic signals as a function of linear polarizer angle ($\varphi$), which are calculated for representative pathways under laser excitation with $\varepsilon_{exc} = 0.2$. Black circles in Fig. 4f show the fifth harmonic signals measured in experiment as a reference. The pathway of $M_x^{inter} + M_x^{intra}$ (orange solid line) generates a linearly-polarized harmonic signal along the x-direction despite the presence of both laser fields $E_x$ and $E_y$, indicating that $I_y^{(5\omega)}$ cannot be generated without $M_y^{inter}$ or $M_y^{intra}$. The pathway of $M_x^{inter} + M_y^{inter}$ (black solid line) generates near linearly-polarized harmonic along a direction slightly rotated from x-direction. Significantly weaker $E_y$, which is five times smaller than $E_x$, generates negligibly small but non-zero $I_y^{(5\omega)}$. However, the pathway of $M_x^{inter} + M_y^{intra}$ (blue solid line) generates surprisingly strong $I_y^{(5\omega)}$ with polarization major axis rotated by 35°. This clearly shows that charge carriers in graphene exhibit the intricate coupling between $M_x^{inter}$ and $M_y^{intra}$ under elliptically polarized excitation. Although calculation of accurate harmonic signals requires full Hamiltonian with all four microscopic channels ($M_x^{inter}$, $M_y^{inter}$, $M_x^{intra}$, and $M_y^{intra}$), it is remarkable to note that the pathway of $M_x^{inter} + M_y^{intra}$ alone reproduces our experiment result (black circles) with a good

agreement. Our theoretical calculation result agrees with recent theoretical results[22] where the strong coupling is attributed to large intraband transition strength along the $k_y$-direction (blue arrow in Fig. 4e ($M_y^{intra}$)) than along the $k_x$-direction (orange arrow in Fig. 4e ($M_x^{intra}$)) for the states excited by interband transition (black arrow in Fig. 4e ($M_x^{inter}$)).

In conclusion, the chemical potential control in our study revealed the quantum pathways of massless Dirac fermions in graphene for HHG under strong laser fields. Charge carriers are excited via the multiphoton interband transitions, which forms destructively interfering quantum pathways for HHG. As the light-matter interaction deviates beyond the perturbative regime, the highly anisotropic carrier excitation profile of massless Dirac fermions in momentum space leads to the intricate coupling between the interband and intraband transitions, which allows to control HHG in terms of intensity and polarization state under elliptically polarized excitation. Considering the fact that the circulating profile of the transition dipole moment (Fig. 4c) universally presents in solids hosting gapless charge carriers with topologically-protected band crossings (see Supplementary note for transition dipole moment of massless Dirac fermions), the underlying principles of HHG in graphene can be also utilized for broad range of

quantum semimetals including topological insulators and Weyl semimetals to resolve quantum pathways of Dirac electrons under strong laser fields. In addition, the intensity and polarization modulation scheme of high harmonics provides a microscopic mechanism to enable a novel light source based on HHG in quantum semimetals with a fast and efficient control via an electrostatic gating.

## Methods

### Device fabrication

Polycrystalline graphene film is grown by chemical vapor deposition on Cu foil (Nilaco corporation, #CU-113213, 30 μm thick, 99.9% purity). The Cu foil is annealed at 1030 °C for 4-hour under flow of $H_2$ at 70 sccm with a total pressure of 5 Torr; then graphene film is grown at 1040 °C for 2-h 30 min by additionally introducing methane in Ar carrier gas at 0.15 sccm. During the cooling process after growth, the methane injection is maintained until the film had cooled to 600 °C, then only $H_2$ is introduced until the temperature reached <150 °C to allow unloading of the sample. Poly (methyl methacrylate) (PMMA, 996 K, 8% in Anisole) is spun onto as grown graphene. To etch the Cu substrate, the PMMA/graphene/Cu foil is floated on a Cu etchant of $FeCl_3$ aqueous solution (Sigma-Aldrich, #667528) for 30 min. After complete etching of Cu, the PMMA/graphene film is transferred onto a surface of ultrahigh purity deionized water (DIW) by scooping with a $SiO_x$ substrate, then and releasing the ultrathin film. The process is repeated twice in a clean DIW to rinse the surface, then the film is transferred onto a double-side polished sapphire substrate. The sample is dried by heating at 60 °C in air for 10 min, followed by annealing at 165 °C for 10 min and soaking in acetone at 60 °C for 10 min to remove the PMMA. Finally, annealing is performed at 350 °C in air for 15 min to remove the PMMA residue.

After transfer process, drain, source, and side gate contacts are deposited by e-beam evaporator (Cr 5 nm/Au 50 nm) to construct ion-gel-based field-effect transistor and distance between drain and source is about a few millimeters. For ion-gel gate dielectric, 0.1 g of poly(-vinylidene fluoride-co-hexafluoropropylene) (PVDF-HFP) and 0.195 g of 1-ethyl-3-methylimidazolium bis (trifluoromethylsulfonyl)imide ([EMIM][TFSI]) are dissolved in the mixture of 2 ml of 2-butanon and 0.25 ml of propylene carbonate. [EMIM][TFSI] and PVDF-HFP were purchased from Sigma Aldrich and ARKEMA, respectively. The solution is stirred overnight at 45 °C, then drop-casted onto the device to cover gate electrodes and graphene channel. The sample is dried in argon filled glove box. Source, drain, and gate contacts are electrically connected by pair of voltage source meter (Keithley 2400, 2450) to apply gate voltage and read channel resistance.

### HHG measurements

Based on femtosecond laser system (Light Conversion PHAROS), mid-infrared pulses are prepared using optical parametric amplifier (ORPHEUS) and difference frequency generator (LYRA). The output serves wavelength-tunable multi-cycle pulses with repetition rate of 100 kHz. The spectral linewidth of the pulse is 15.4 meV in full-width half-maximum and the pulse duration is estimated to be 120 fs assuming a Fourier-transform-limited pulse. To control its ellipticity, liquid crystal retarder (Thorlabs LCC1111-MIR) is employed, whose optical axis is oriented at an angle of 45° with respect to the laser polarization. Then, mid-infrared pulses are focused at roughly center of the graphene device by ZnSe focusing objectives with spot size of 150 μm. Emitted HHG has been collected by 50X objective lens on transmission geometry, and its polarization is analyzed by half-wave plate mounted on motorized stage and fixed Glan-Taylor polarizer. The HHG spectra are recorded by an electron-multiplying charge-coupled device detector (ProEM, Princeton instruments) and grating spectrometer (SP-2300, Princeton instruments) at Materials Imaging & Analysis Center of POSTECH.

## Data availability

All of the data that support the findings of this study are available in the main text or Supplementary Information. Source data are available from the corresponding authors on request.

## Code availability

Custom codes used in this work can be provided by the corresponding author on request.

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

## Acknowledgements

J.K. and M.K. acknowledge help from D.P. and Prof. U.J. for graphene device preparation. S.C., M.K., M.-H.J., and J.K. acknowledge the support from Institute of Basic Science (IBS-R034-D1). J.K. and M.K. acknowledge the support from the National Research Foundation of Korea grants (NRF-2020R1A4A1018935, and 2020R1A2C2103166). This study was supported by Brain Korea 21 FOUR project for Education and research center for future materials (F21YY7105002). This research was also supported by the MSIT (Ministry of Science and ICT), Korea, under the ITRC (Information Technology Research Center) support program (IITP-2022-RS-2022-00164799) supervised by the IITP (Institute for Information & Communications Technology Planning & Evaluation). Y.K. and J.L. acknowledge the support from the DGIST R&D program (22-CoE-NT-01). J.K. and J.L. also acknowledge the support from the national research foundation (NRF) of Korea (2022R1A2B5B01001582). G.Y.C. acknowledges the support of the National Research Foundation of Korea (NRF) funded by the Korean Government (Grant No. 2020R1C1C1006048) and Grant No. IBS-R014-D1. G.Y.C. acknowledges the Air Force Office of Scientific Research under Award No. FA2386-20-1-4029 and No. FA2386-22-1-4061. G.Y.C. also thanks the support by Samsung Science and Technology Foundation under Project Number SSTF-BA2002-05. C.-J.K. acknowledges the support from the National Research Foundation of Korea grants (2020R1C1C1014590 and 2022M3H4A1A01012718).

## Author contributions

J.K. conceived the project. S.C. and M.K. built optical set-up and obtained HHG spectra. S.C., M.K., S.C., and S.K. fabricated graphene devices with a gate control. Y.K., G.C., and H.K. obtained theoretical calculation results of HHG in graphene. C.-J.K., M.J.P., S.C., M.K., S.C., and S.K. analyzed gating behavior of graphene with an ion-gel under strong laser field. J.K., M.-H.J., J.D.L., B.J.K., G.C., S.C., M.K., Y.K., and H.K. analyzed HHG data. All authors discussed and wrote the manuscript together.

## Competing interests

The authors declare no competing interests.
