## [Peer Review File · Nature Communications]

Reviewers' Comments:

Reviewer #1:

Remarks to the Author:

P The manuscript entitled "Imaging quantum pathways of lightwave-driven massless Dirac fermions" by Soonyoung Cha et al. presents an interesting work on high harmonic generation (HHG) from graphene embedded in a field-effect transistor device. The authors measured HHG intensity and polarization as a function of hole doping level, driven by linearly and elliptically polarized mid-infrared laser pulses. They showed that multiphoton excitation channels are destructively interfered and can be partially closed by tuning the chemical potential. In addition, they found that the complex dynamics between the interband and intraband transitions can be decoupled, according to the dependence of HHG polarization states on the laser ellipticity. The authors suggested that quantum pathways of massless Dirac fermions in graphene can be mapped out through HHG spectroscopy.

The manuscript reports a beautiful experiment, but I have several questions and concerns on the viewpoints.

1. The authors claimed that they observed the lightwave-driven dynamics of massless Dirac fermions, but the laser intensity of 3.1 GW/cm² was adopted in measurement shown in Fig. 2-3. Unfortunately it is not in the non-perturbative regime. As displayed in Fig. 1e, the intensity of the 5th harmonic scales as $I_5 \propto I^5$ @ $I = 3.1 \text{ GW/cm}^2$, in which multiphoton excitation dominates the nonlinear current and gives perturbative harmonic generation. The authors may check if the experimental observation can be reproduced in the framework of nonlinear optics by taking into account the fifth-order susceptibility.

In the intense laser field regime, strong field ionization, such as Zener tunneling [Science 356, 736 (2017); Nature 550, 224 (2017)], the subsequent acceleration of electron-hole pair and generalized recollision lead to the emission of high harmonics [Phys. Rev. Lett. 113, 073901 (2014)]. In this work the laser field strength may not be the primary driving force of Dirac electron dynamics, but the multi-photon transitions.

The enhancement of the 5th harmonic, H₅, is attribute to the partial closing of competing transition channels due to Pauli blocking. To this extent, the closing of interference channels dose not possess lightwave-driven character. Therefore, whether the present finding can be called as lightwave-driven Dirac fermions is open to debate, and the authors should give more convincing arguments on this.

2. In Fig. 2c and d, there exists a dip around $|\mu| = 0.4 \text{ eV}$ on the curve of I_5 versus chemical potential. Why is that? Should this phenomenon be attribute to selectively closing of some interference channels, or the jitter of the measurements? Also, note that the dip is absent in Fig. S4b, please clarify.

3. The number of photons involved in multiphoton processes should be an integer. It is interesting to notice that the resonance-like profiles of I_5 peaks at $|\mu| = 3.4$. Intuitively,

the Pauli blocking should take place once $|\mu|$ exceeds 3ϕ . Could the authors give an explanation?

4. Is there a connection between the linear energy dispersion of the Dirac cone and the observed modulation of HHG? If the Dirac band replaced by other types of dispersion, such as parabolic band with a narrower bandgap, can the unique ellipticity dependence of HHG still be observed?

5. Some key experimental parameters are not given in the manuscript, such as, dimensions of the graphene sample between the electrodes, and the focal spot size of the pump pulses. Will the H₅ be affected if the focused laser illuminates on the graphene-metal junction? How to exclude that influence?

6. Figure 1e shows that the pump intensity of $\approx 3.1 \text{ GW/cm}^2$ can be viewed as a starting point where electron dynamics changes from perturbative to non-perturbative. This transition region has not been well explored previously, and the competition between the multiphoton excitation and lightwave-driven electron dynamics ought to be revealed. I would be happy to see it being clarified.

Reviewer #2:

Remarks to the Author:

In their work, Cha et al. present an interesting experimental work on high-harmonic generation (HHG) in graphene monolayer, supported by theoretical simulations. They investigate the role of chemical doping on the HHG in graphene and demonstrate interesting experimental findings, in particular the role of the chemical doping on the polarization state of the emitted harmonics. Because of the specific nature of the Dirac cone, the authors show that the interband excitation channel can be suppressed in the strong doping case, thus affecting the harmonic emission. This work is timely and interesting and deserves publication. However, I find that the authors must address some points before the manuscript can be recommended for publication. These points are listed below:

- i) I find the title over-claiming the results of the manuscript. What the authors show is at best a dependence on the pathways. I felt to see in which aspect they are visualizing them. Especially, the understanding of the results can only be made because of the theoretical support, and the experiment alone cannot "visualize" anything.
- ii) I think that the explanation of the forbidden excitation channel is related to some prior works. In particular, in condensed matter, there is a quantity called the joint-density-of-state, which is exactly related to the quantity of available optical transitions. Maybe the authors could refer to this quantity and compute it for the different doping, as done for instance in [Phys. Rev. Lett. 118, 087403 (2017)] or in [Nature Photonics volume 14, pages 183–187 (2020)].
- iii) While the results are interesting, I find the analysis of the results a bit weak, especially for Fig. 2. Following the argument in introduction, one would expect an abrupt change of the yield versus doping at exactly $2\mu/E_{ph}=1$. However, around this value, almost nothing is happening. I think that the authors should explain this better.
- iv) The same figure displays a maximum around 3.4, as nicely shown by the authors for various samples. However, again, this value is not explained. At the moment, we do not know if this value is due to a material's property, a laser property, or a combination of both. I think that the simulations here could come at help. The authors could also simulate the results of Fig. 2 and try to explain the physical origin of the maximum at 3.4. Which mechanism is dominant for instance. In fact, the same plot as Fig. 2c-d, but decomposed into interband and intraband could really provide some insights here. I would suggest the authors to add this simulations.

1. Reply to Reviewer #1

Original comment (1):

General remarks of Reviewer 1:

The manuscript entitled “Imaging quantum pathways of lightwave-driven massless Dirac fermions” by Soonyoung Cha et al. presents an interesting work on high harmonic generation (HHG) from graphene embedded in a field-effect transistor device. The authors measured HHG intensity and polarization as a function of hole doping level, driven by linearly and elliptically polarized mid-infrared laser pulses. They showed that multiphoton excitation channels are destructively interfered and can be partially closed by tuning the chemical potential. In addition, they found that the complex dynamics between the interband and intraband transitions can be decoupled, according to the dependence of HHG polarization states on the laser ellipticity. The authors suggested that quantum pathways of massless Dirac fermions in graphene can be mapped out through HHG spectroscopy. The manuscript reports a beautiful experiment, but I have several questions and concerns on the viewpoints.

Our reply:

We deeply appreciate the referee for her/his effort to review our manuscript. In addition, we thank for her/his appreciation of the importance of our work and valuable feedbacks which have been tremendously helpful to improve our manuscript. We have addressed her/his questions and concerns carefully as below.

Original comment (2):

1. The authors claimed that they observed the lightwave-driven dynamics of massless Dirac fermions, but the laser intensity of 3.1 GW/cm^2 was adopted in measurement shown in Fig. 2-3. Unfortunately it is not in the non-perturbative regime. As displayed in Fig. 1e, the intensity of the 5th harmonic scales as $I_H^{5\omega} \propto I_{exc}^5 @ I_{exc} = 3.1 \text{ GW/cm}^2$, in which multiphoton excitation dominates the nonlinear current and gives perturbative harmonic generation. The authors may check if the experimental observation can be reproduced in the framework of nonlinear optics by taking into account the fifth-order susceptibility. In

the intense laser field regime, strong field ionization, such as Zenner tunneling [Science 356, 736 (2017); Nature 550, 224 (2017)], the subsequent acceleration of electron hole pair and generalized recollision lead to the emission of high harmonics [Phys. Rev. Lett. 113, 073901 (2014)]. In this work the laser field strength may not be the primary driving force of Dirac electron dynamics, but the multi-photon transitions. The enhancement of the 5th harmonic, H5, is attribute to the partial closing of competing transition channels due to Pauli blocking. To this extent, the closing of interference channels does not possess lightwave-driven character. Therefore, whether the present finding can be called as lightwave-driven Dirac fermions is open to debate, and the authors should give more convincing arguments on this.

Our reply:

We appreciate the reviewer for her/his critical comment on the interpretation of our experimental results. We totally agree with the reviewer that enhancement of $I_H^{5\omega}$ through multi-photon transitions can be well-explained by light-matter interaction in the perturbative regime. However, although it is not completely in the non-perturbative regime, our results at 3.1 GW cm⁻² reports emerging new features in HHG which cannot be explained by the perturbative electron dynamics.

First of all, our apology for the unclear presentation of Fig. 1e in the previous version. For the laser intensity of 3.1 GW cm⁻² and its neighboring intensity range, the 5th harmonic intensity roughly scales as $I_H^{5\omega} \propto I_{exc}^{\sim 3.8}$ (Fig. R1), which indicates the light-matter interaction is deviating from the perturbative regime. We have added a black arrow to indicate the specific data point clearly in the log-log plot in the revised manuscript.

In addition, under elliptically polarized excitation, the polarization axis of the 5th harmonics rotates as the laser intensity increases from 2.4 GW cm^{-2} to 10.1 GW cm^{-2} (Fig. 1g) while the ellipticity of the laser pulse is fixed at 0.3. This rotation of the polarization axis is clear indication of the non-perturbative response at 3.1 GW cm^{-2} . In nonlinear optics in the perturbative regime, where the fifth-order susceptibility are independent on the laser intensity, the polarization axis cannot rotate as the intensity increases. We assume that the perturbative contribution from higher-order susceptibility on the fifth harmonic generation is negligible considering more than an order magnitude weaker 7th harmonic generation as shown in Fig. 1d which is taken even at 50 GW cm^{-2} .

In the perturbative nonlinear optics, the intensity of a fifth harmonic signal $I_i(5\omega)$ under laser-field $E(\omega)$ can generally be calculated by the following equation

$$I_i(5\omega) \propto \left| P_i^{(5)}(5\omega) \right|^2 \propto \left| \sum_{jklmn} \chi_{ijklmn}^{(5)}(5\omega) E_j(\omega) E_k(\omega) E_l(\omega) E_m(\omega) E_n(\omega) \right|^2$$

where $P_i^{(5)}(5\omega)$ is a induced nonlinear polarization at 5ω and $\chi_{ijklmn}^{(5)}$ is the component of fifth-order nonlinear susceptibility tensor. In our HHG set up in the transmission geometry, E_z becomes zero, which simplifies all components as follows

$$\begin{pmatrix} \tilde{P}_x \\ \tilde{P}_y \end{pmatrix} = \begin{pmatrix} \chi_{xxxxxx} & \chi_{xyyyyy} & \chi_{xxxyyy} & \chi_{xxxxyy} & \chi_{xxxxxy} & \chi_{xxxxxy} \\ \chi_{yxxxxx} & \chi_{yyyyyy} & \chi_{yxxyyy} & \chi_{yxxxyy} & \chi_{yxxxxy} & \chi_{yxxxxy} \end{pmatrix} \begin{pmatrix} E_x^5 \\ E_y^5 \\ E_x E_y^4 \\ E_x^2 E_y^3 \\ E_x^3 E_y^2 \\ E_x^4 E_y \end{pmatrix}.$$

\tilde{P}_x and \tilde{P}_y are the fifth harmonic polarization along x-axis and y-axis, respectively, which can be expressed with the following equations

$$\begin{aligned} \tilde{P}_x &= P_x e^{i\delta_x} \\ \tilde{P}_y &= P_y e^{i\delta_y} \end{aligned}$$

where $P_x(P_y)$ and $\delta_x(\delta_y)$ are magnitude and phase of the polarization along x-axis (y-axis), respectively. In the case of the laser excitation of ellipticity ϵ , E_y can be replaced to $(i\epsilon)E_x$

$$\begin{pmatrix} \tilde{P}_x \\ \tilde{P}_y \end{pmatrix} = \begin{pmatrix} \chi_{xxxxxx} & \chi_{xyyyyy} & \chi_{xxxyyy} & \chi_{xxxxyy} & \chi_{xxxxxy} & \chi_{xxxxxy} \\ \chi_{yxxxxx} & \chi_{yyyyyy} & \chi_{yxxyyy} & \chi_{yxxxyy} & \chi_{yxxxxy} & \chi_{yxxxxy} \end{pmatrix} \begin{pmatrix} E_x^5 \\ (i\epsilon)^5 E_x^5 \\ (i\epsilon)^4 E_x^5 \\ (i\epsilon)^3 E_x^5 \\ (i\epsilon)^2 E_x^5 \\ (i\epsilon)^1 E_x^5 \end{pmatrix}$$

\tilde{P}_x and \tilde{P}_y are represented by tensor components and E_x

$$\begin{aligned} \tilde{P}_x &= (\chi_{xxxxxx} + \chi_{xyyyyy}(i\epsilon)^5 + \chi_{xxxyyy}(i\epsilon)^4 + \chi_{xxxxyy}(i\epsilon)^3 + \chi_{xxxxxy}(i\epsilon)^2 + \\ &\quad \chi_{xxxxxy}(i\epsilon)) E_x^5. \\ \tilde{P}_y &= (\chi_{yxxxxx} + \chi_{yyyyyy}(i\epsilon)^5 + \chi_{yxxyyy}(i\epsilon)^4 + \chi_{yxxxyy}(i\epsilon)^3 + \chi_{yxxxxy}(i\epsilon)^2 + \\ &\quad \chi_{yxxxxy}(i\epsilon)) E_x^5. \end{aligned}$$

Therefore, both relative magnitude and phase of \tilde{P}_x and \tilde{P}_y do not change over laser intensity. For elliptical polarization, angle of the major polarization axis θ can be expressed with P_x , P_y , and relative phase δ between P_x and P_y ($\delta = \delta_x - \delta_y$) [Hecht, E. *Optics* (Pearson, 2017)]

$$\tan 2\theta = \frac{2P_x P_y \cos\delta}{(P_x)^2 - (P_y)^2} = \frac{2(P_x/P_y) \cos\delta}{(P_x/P_y)^2 - 1}$$

Thus, this indicates that the polarization axis of high harmonics cannot rotate over intensity in the perturbative regime.

More interestingly, in Fig. 3b and 3c, rotation direction of the polarization axis (either clockwise or counter-clockwise) can be controlled by the helicity of the elliptically polarized laser (i.e. the relative phase of the laser fields along the x- and y-directions).

Therefore, we believe that the non-perturbative electron dynamics significantly contribute to our experimental data (Fig. 2 and 3) measured at 3.1 GW cm^{-2} although the perturbative electron dynamics such as the multi-photon transitions contributes strongly as well. However, we totally agree with the reviewer that more rigorous studies including systematic control of carrier envelop phase (CEP) of the laser field are required to claim our observed phenomena as the lightwave-driven phenomena. We have removed our expression “lightwave-driven” referring to our observed phenomena in graphene and included the discussion above in the revised main text and supplementary information following the reviewer’s comment.

Original comment (3):

2. In Fig. 2c and d, there exists a dip around $2|\mu| = 0.4 \text{ eV}$ on the curve of $I_x^{5\omega}$ versus chemical potential. Why is that? Should this phenomenon be attribute to selectively closing of some interference channels, or the jitter of the measurements? Also, note that the dip is absent in Fig. S4b, please clarify.

Our reply:

We thank the reviewer for her/his effort to carefully check our experimental data and provide us valuable comments which encourage us to obtain deeper understanding on the chemical potential dependence of the 5th harmonic signal. In fact, experimental data in Fig. 2 and Fig. S4 are taken from two different samples. Despite being taken under the similar laser intensity, the samples exhibit slightly different features including the presence of a

dip-like profile around $2|\mu| = 0.4$ eV as the reviewer points out. We observe that the dip-like profiles are also occasionally observed in other samples (Fig. R2). The dip-like profiles around $2|\mu| = 0.4$ eV shown in Fig. 2c is observed in sample 1 and 2, but not in sample 3 and 4. This implies that quality of ion-gel or graphene grown by chemical vapor deposition (CVD) might affect the detailed features in the curve of $I_x^{5\omega}$ versus chemical potential.

Interestingly, our theoretical calculation (Fig. R3) also shows a dip-like profile around at similar chemical potential. As dephasing time T_2 decreases from 4 fs to 2 fs, the dip-like profile becomes more pronounced while the resonance-like profile (originating from channels via multiphoton transitions) is heavily suppressed. In fact, the theoretical calculation reveals that two characteristic profiles are combined in $I_x^{5\omega}$ as a function of chemical potential: the resonance-like profile at $2|\mu|/E_{ph} = 3.4$ and the plateau-like profile at $2|\mu|/E_{ph} < 2$. While the resonance-like profile is originated from multi-photon transition channels, the physical origin of the plateau-like profile is not clear at the moment. Identifying the physical origin of this profile requires comprehensive experimental and theoretical investigation with systematic control of laser intensity and sample qualities

which will allow to quantify and to control dephasing time. Although this will be certainly of our future interest, unfortunately, this goes beyond the scope of the current study. We have included the discussion above in the main text and the supplementary information of the revised manuscript.

Original comment (4):

3. The number of photons involved in multiphoton processes should be an integer. It is interesting to notice that the resonance-like profiles of $I_x^{5\omega}$ peaks at $\frac{2|\mu|}{E_{ph}} = 3.4$. Intuitively, the Pauli blocking should take place once $2|\mu|$ exceeds $3E_{ph}$. Could the authors give an explanation?

Our reply:

We appreciate the reviewer for raising this important question on the resonance-like profile of $I_x^{5\omega}$ which shows the maximum intensity at $2|\mu|/E_{ph} = \text{non-integer}$. As the reviewer points out, the number of photons involved in multiphoton transition processes is an integer which will show series of multiple sharp resonance-like profiles at $2|\mu|/E_{ph} = n$ (n is an integer). In the perturbative regime without considering the scattering process, such profile has been theoretically predicted for $I_x^{5\omega}$ by the recent theoretical study [PRB 99, 195407(2019)], which we have drawn in Fig. R4 using their results.

Figure. R4. Perturbative theoretical calculation of $\chi^{(5)}$ without considering scattering process. The fifth harmonic susceptibility of graphene $\chi^{(5)}$ versus $2|\mu|/E_{ph}$ is calculated without any broadening contributions, exhibiting series of sharp resonance-like profiles at $2|\mu|/E_{ph} = n$.

However, intense laser excitation creates high density of photocarriers in graphene, which enables rapid electron scattering process. Then, series of sharp resonance-like profiles can be drastically broadened and merged together, forming one resonance-like profile located at $2|\mu|/E_{ph} = \text{non-integer}$. For example, even in the perturbation regime, the recent experimental and theoretical results [Fig. 4c in Nature Photonics **12**, 430–436 (2018) and New J. Phys. **16**, 053014 (2014)] shows that the 3rd harmonic intensity becomes the highest when $2|\mu|/E_{ph}$ is away from 2.

In order to investigate such possibility for $I_x^{5\omega}$ in our case, we have theoretically calculated 5th harmonic intensity as a function of chemical potential (Fig. R5) by employing quantum master equation which fully takes into account electronic scattering processes under strong laser field. Fig. R5b and R5c show how massless Dirac states around the Dirac point in the momentum space contributes to generate 5th harmonic current $J_x^{5\omega}$ in graphene under linearly polarized excitation along the x-direction (E_x). Dashed lines are constant energy contours on a Dirac cone, which describes electronic states vertically separated by energy mE_{ph} . m is an integer. Real and imaginary parts of $J_x^{5\omega}$ (Fig. R5b and R5c, respectively) are most strongly generated from Dirac states in the region between $2E_{ph}$ and $4E_{ph}$. Depending on the photon number required for multi-photon transitions and the azimuthal angle, Dirac states generate $J_x^{5\omega}$ with characteristic sign and magnitude, which destructively interfere when radiating $I_x^{5\omega}$. As $2|\mu|$ increases, Pauli blocking sequentially disables contribution from the Dirac states starting from near the Dirac point to higher energy. As shown in Fig. R5a, $I_x^{5\omega}$ increases as the destructive interference is partly eliminated while $I_x^{5\omega}$ eventually disappears as all the resonant Dirac states are disabled. Our theoretical calculation (Fig. R5a) also shows that the maximum intensity is exhibited around at $2|\mu|/E_{ph} = 3.4$, which is the most common value for $2|\mu|/E_{ph}$ in our experimental data (Fig.S5). We have included the discussion above in the revised manuscript.

Figures. R5. Theoretical calculation of the fifth harmonic intensity and current amplitude in the momentum space. **a**, Calculated fifth harmonic intensity as a function of $2|\mu|/E_{ph}$ under linearly-polarized excitation with I_{exc} of 3.1 GW cm^{-2} . **b and c**, (b) Real part and (c) imaginary part of 5th harmonic current $J_x^{5\omega}$ generated by Dirac states around Dirac point in the momentum space. The red (blue) color represents for positive (negative) values. Dashed lines are constant energy contours on a Dirac cone, which describe the conduction band states vertically separated by nE_{ph} from the valence band.

Original comment (5):

4. Is there a connection between the linear energy dispersion of the Dirac cone and the observed modulation of HHG? If the Dirac band replaced by other types of dispersion, such as parabolic band with a narrower bandgap, can the unique ellipticity dependence of HHG still be observed?

Our reply:

We appreciate the reviewer for her/his critical question on a connection between the linear energy dispersion and the ellipticity dependence of HHG in graphene. In the regime where carriers are dominantly excited by multi-photon transitions, our studies show that the ellipticity dependence of HHG originates from the nonlinear coupling between interband and intraband transitions. For the linear energy band dispersion from $H = \pm v_F \sigma \cdot \mathbf{k}$, linearly polarized laser-field along the x-direction (E_x) excites charge carriers in the peculiar anisotropic distribution in the momentum space as shown in Fig. 4a and 4b, which reflects the transition dipole moment texture of massless Dirac fermion (Fig. 4c). Under elliptically polarized laser-field, additional laser-field along the y-direction with a phase delay of $\pi/2$ (E_y) can modify the anisotropic distribution by driving charge carriers within the same bands. Our theoretical calculation (Fig. 4f) shows that this pathway ($M_x^{inter} + M_y^{intra}$) can generate I_y efficiently.

In order to answer the reviewer’s question, we have investigated the ellipticity dependence of HHG from the “massive” Dirac fermions by introducing a gap of ~ 600 meV which corresponds to $\sim 2.2E_{ph}$. E_{ph} is the photon energy of laser excitation in our experiment. Both the linear band dispersion (black solid line in Fig. R6c) and the anisotropic transition dipole moment texture (Fig. R6d) are relaxed in the cases of “massive” Dirac fermions (blue solid line in Fig. R6c and Fig. R6e). Under the same laser intensity and ellipticity, massive Dirac fermions generates significantly smaller I_y (red solid line in Fig. R6b) in comparison to massless Dirac fermions (red solid line in Fig. R6a). This indicates that the observed ellipticity dependence of HHG is strongly connected to the linear band dispersion. However, we note that considerable amount of I_y can be still generated even from massive Dirac fermions. This is potentially because the anisotropy of the transition dipole moment texture (Fig. R6e) is not completely relaxed.

Figure R6. Comparison between massless Dirac fermions and massive Dirac fermions. a and b, Calculated fifth harmonic intensity as a function of ϵ_{exc} on massless

Dirac Hamiltonian with $2|\mu| = 2.2E_{\text{ph}}$ (a) and massive Dirac Hamiltonian with $E_g = 2.2E_{\text{ph}}$ (b). **c**, The Band dispersion of massless Dirac fermions (black curve) which is linear and massive Dirac fermion (blue curve). **d and e**, Transition dipole texture of massless (d) and massive (e) Dirac dispersion.

In principle, we suspect that similar ellipticity dependence of HHG might be observable from parabolic bands with a narrow bandgap if their transition dipole moment exhibits strong anisotropy like massless Dirac fermions. However, detailed information of wavefunctions and crystal structures is required in order to search for such a system, which is unfortunately beyond the scope of the current work. On the other hand, in stronger field regime where charge carriers are driven in light-induced semi-metallized band structure under extremely strong laser-field, the previous work [Phys. Rev. B 94, 241107 (2016)] shows that even parabolic bands can efficiently generate I_y under elliptically polarized excitation.

Therefore, in the revised manuscript, we have changed our expression “unique mechanism for massless Dirac fermions” to “characteristic mechanism for massless Dirac fermions” in order to address the reviewer’s concern.

Original comment (6):

5. Some key experimental parameters are not given in the manuscript, such as, dimensions of the graphene sample between the electrodes, and the focal spot size of the pump pulses. Will the H5 be affected if the focused laser illuminates on **the graphene–metal junction**? How to exclude that influence?

Our reply:

We appreciate the reviewer for her/his effort to carefully check our experimental configuration. Following the reviewer’s comments, we have included information about dimension of the graphene samples and the focal spot size of pump pulses in the revised

manuscript. The mid-infrared pulse is focused on graphene by a ZnSe objective lens, and the spot size of the pulse is measured as $150 \mu\text{m}$ (full width at half-maximum of intensity, FWHM) by using knife-edge method. Our graphene device is shown in the Fig. R7 below. The size of graphene device is typically a few millimeters by a few millimeters, which is wide enough to avoid the graphene-metal junction being illuminated by focused laser. During the experiment, the mid-infrared laser beam is illuminated roughly at the center of graphene devices by monitoring position of a guiding HeNe laser (632 nm) beam spot which propagates along the same optical path of mid-infrared laser.

Figure. R7. Image of the graphene device on the slide glass. Graphene is located between the source and drain contact, surrounded by red lines.

Original comment (7):

6. Figure 1e shows that the pump intensity of $I_{\text{exc}} = 3.1 \text{ GW/cm}^2$ can be viewed as a starting point where electron dynamics changes from perturbative to non-perturbative. This transition region has not been well explored previously, and the competition between the multiphoton excitation and lightwave-driven electron dynamics ought to be revealed. I would be happy to see it being clarified.

Our reply:

We appreciate the reviewer for providing us an invaluable comment that our pump intensity can be considered as a transitioning point where electron dynamics change from perturbative to non-perturbative. In the perturbative regime, generation of 5th harmonic can be explained mostly by resonances from multi-photon interband transitions. However, as the laser intensity increases, high harmonics starts showing distinctive features which cannot be explained by the perturbative electron dynamics. At $I_{\text{exc}} = 3.1 \text{ GW cm}^{-2}$, charge carriers in graphene are still dominantly excited by multiphoton interband transitions while a nonlinear coupling with intraband transitions plays an important role to generate high harmonics. Multi-photon transitions excite charge carriers in a specific distribution in the momentum space, which is subsequently driven by intraband dynamics. In graphene, this coupling enables efficient generation of high harmonics under elliptically polarized laser-field, which drastically rotates polarization axis of 5th harmonics following the helicity of the laser pump.

As the laser intensity increases further, we would expect that charge carriers are excited dominantly by Zener tunneling [*Nature* **550**, 224–228 (2017), *Phys. Rev. B* **94**, 241107 (2016), *Phys. Rev. Lett.* **116**, 016601 (2016)] or light-induced semi-metallized band structure [*Rev. B* **94**, 241107 (2016), *Phys. Rev. Lett.* **116**, 016601 (2016)]. Chemical potential dependence of HHG at higher laser intensity will allow to investigate how charge carriers excited by different mechanisms interfere each other. Unfortunately, such investigation is experimentally challenging at this moment due to the long-term laser damage issue as shown in Fig. S2. HHG study with fewer cycle laser pulses can help to address the laser damage issue, which is of our future interest. We have included the discussion above in the introduction and conclusion parts of the revised manuscript.

2. Reply to Reviewer #2**Original comment (1):**

In their work, Cha et al. present an interesting experimental work on high-harmonic generation (HHG) in graphene monolayer, supported by theoretical simulations. They investigate the role of chemical doping on the HHG in graphene and demonstrate

interesting experimental findings, in particular the role of the chemical doping on the polarization state of the emitted harmonics. Because of the specific nature of the Dirac cone, the authors show that the interband excitation channel can be suppressed in the strong doping case, thus affecting the harmonic emission. This work is timely and interesting and deserves publication. However, I find that the authors must address some points before the manuscript can be recommended for publication. These points are listed below:

Our reply: We deeply appreciate the referee for her/his effort to review our manuscript. In addition, we thank for her/his appreciation of the importance of our work and valuable comments on the experimental and theoretical results which have been tremendously helpful to improve our manuscript. We have addressed her/his questions and concerns carefully as below.

Original comment (2):

1. I find the title over-claiming the results of the manuscript. What the authors show is at best a dependence on the pathways. I felt to see in which aspect they are visualizing them. Especially, the understanding of the results can only be made because of the theoretical support, and the experiment alone cannot "visualize" anything.

Our reply:

We appreciate the reviewer for her/his critical comment on the title of our manuscript. We agree with the reviewer that our expressions such as "imaging" or "visualizing" would not be appropriate. Although our experimental results combined with the theoretical support provide critical information to resolve pathways of high harmonic generation in graphene, we do not physically image their trajectory either in the momentum space or the real space. Following the reviewer's comment, we have changed the title of our manuscript to 'Gate-tunable quantum pathways of high harmonic generation in graphene' and revised all the related descriptions in the main text.

Original comment (3):

2. I think that the explanation of the forbidden excitation channel is related to some prior works. In particular, in condensed matter, there is a quantity called the joint-density-of-state, which is exactly related to the quantity of available optical transitions. Maybe the authors could refer to this quantity and compute it for the different doping, as done for instance in [Phys. Rev. Lett. 118, 087403 (2017)] or in [Nature Photonics volume 14, pages 183–187 (2020)].

Our reply:

We appreciate for the reviewer's suggestion to improve our explanation on the observed phenomena by using a quantity such as the joint-density-of-states (JDOS). For linear optical process, JDOS is simply proportional to E from the Dirac point in graphene [J. Phys. Soc. Jpn. 75, 074716 (2006)]. However, for high-order nonlinear optical processes like 5th harmonic generation in our study, calculating JDOS can be quite complicated due large number of intermediate states involved in high-order electronic processes. Furthermore, even for given energy separation (E) between valence and conduction bands, electronic states can generate nonlinear current differently depending on their exact azimuthal positions in the momentum space due to transition dipole moment, for example, as shown in Fig.4a and 4b.

In order to address the reviewer's comment, we have theoretically calculated map of $J_x^{5\omega}$ in the momentum space (Fig. R1b and R1c) which describes how massless Dirac states contributes to generate 5th harmonic current $J_x^{5\omega}$ in graphene under linearly polarized excitation along the x-direction (E_x). Dashed lines are constant energy contours on a Dirac cone, which describes electronic states vertically separated by energy mE_{ph} (m is an integer). Real and imaginary parts of $J_x^{5\omega}$ (Fig. R1b and R1c, respectively) are most strongly generated from Dirac states in the region between $2E_{ph}$ and $4E_{ph}$. Depending on the photon number required for multi-photon transitions and the azimuthal angle, Dirac states generate $J_x^{5\omega}$ with characteristic sign and magnitude, which destructively interfere when radiating $I_x^{5\omega}$. As $2|\mu|$ increases, Pauli blocking sequentially disables contribution

from the Dirac states starting from near the Dirac point to higher energy. As shown in Fig. R1a, $I_x^{5\omega}$ increases as the destructive interference is partly eliminated while $I_x^{5\omega}$ eventually disappears as all the resonant Dirac states are disabled. Our theoretically calculated result (Fig. R1a) shows excellent agreement with our experimental data (Fig. 2c). Following the reviewer's comment, we have included the discussion above in the revised manuscript in order to improve our explanation on how forbidden channels contribute to HHG in graphene.

Original comment (4):

3. While the results are interesting, I find the analysis of the results a bit weak, especially for Fig. 2. Following the argument in introduction, one would expect an abrupt change of the yield versus doping at exactly $2\mu/E_{ph}=1$. However, around this value, almost nothing is happening. I think that the authors should explain this better.

Our reply:

We are grateful for the reviewer' suggestion to improve the analysis of our experimental results on chemical potential dependence of 5th harmonics intensity. As the reviewer points out, one photon transition does excite tremendous number of charge carriers, which can be blocked when $2|\mu|/E_{ph} = 1$. However, only small fraction of these photo-excited carriers participates to generate 5th harmonics through series of interband or intraband transitions across all electronic states in the momentum space. Instead, carrier excitation via multi-photon transitions can provide more efficient pathways to generate 5th harmonics as the recent theoretical study [PRB 99, 195407(2019)] demonstrates. In the perturbation regime, Fig. R2 shows susceptibility for 5th harmonics generation as a function of chemical potential. Contribution from one photon transition is drastically weaker than three- or four-photon transitions.

Figure. R2. Perturbative theoretical calculation of $\chi^{(5)}$ without considering scattering process. The fifth harmonic susceptibility of graphene $\chi^{(5)}$ versus $2|\mu|/E_{ph}$ is calculated without any broadening contributions, exhibiting series of sharp resonance-like profiles at $2|\mu|/E_{ph} = n$.

Furthermore, as schematically shown in Fig. R3, our mid-infrared excitation laser has many-cycles of laser-field. Before reaching sufficiently strong laser-field in the middle of the pulse, where supposedly the 5th harmonic is most strongly generated, first few cycles of laser-field can create large number of background photocarriers which effectively increase doping in graphene even when “static” chemical potential is placed at the Dirac point. For this reason, we believe that we do not observe any abrupt change at $2|\mu|/E_{ph} = 1$. We have included the discussion above in the revised manuscript.

Original comment (5):

4. The same figure displays a maximum around 3.4, as nicely shown by the authors for various samples. However, again, this value is not explained. At the moment, we do not know if this value is due to a material’s property, a laser property, or a combination of both. I think that the simulations here could come at help. The authors could also simulate the results of Fig. 2 and try to explain the physical origin of the maximum at 3.4. Which mechanism is dominant for instance. In fact, the same plot as Fig. 2c-d, but decomposed

into interband and intraband could really provide some insights here. I would suggest the authors to add this simulations.

Our reply:

We appreciate the reviewer for this important comment which encourages us to provide deeper understanding on 5th harmonic generation, which is tremendously helpful to improve our manuscript. Following the reviewer's suggestion, we have theoretically calculated $J_x^{5\omega}$ from Dirac states in the momentum space which are decomposed to separate contributions from interband and intraband transitions (Fig. R4). Both interband and intraband transitions participate actively in 5th harmonic generation. Intensity of 5th harmonic generation is determined by summation of $J_x^{5\omega}$ from all the Dirac states. Fig. R4 shows intensity of 5th harmonic generation as a function of chemical potential (black solid line). The resonance-like profile around at $2|\mu|/E_{ph} = 3.4$ is largely originated from interband transitions (red solid line) instead of intraband transition (blue solid line).

Figure. R4. Calculated harmonic intensity as a function of $2|\mu|/\hbar E_{ph}$. Harmonic intensity is decomposed into harmonics from interband transition, I_{inter} (red curve) and harmonics from intraband transition, I_{intra} (blue curve). I_{inter} and I_{intra} are obtained from the $J_{inter}(\tau) = \sum_{\mathbf{k}} \text{Tr} \left[\frac{\partial H_{\mathbf{k}}(\tau)}{\partial \mathbf{k}} \rho_{\mathbf{k}}^{(\text{off-diag})}(\tau) \right]$, and the $J_{intra}(\tau) = \sum_{\mathbf{k}} \text{Tr} \left[\frac{\partial H_{\mathbf{k}}(\tau)}{\partial \mathbf{k}} \rho_{\mathbf{k}}^{(\text{diag})}(\tau) \right]$, respectively. The full harmonic emission, I_{tot} , is given by,

$\mathbf{J}_{\text{tot}}(\tau) = \mathbf{J}(\tau) = \sum_{\mathbf{k}} \text{Tr} \left[\frac{\partial H_{\mathbf{k}}(\tau)}{\partial \mathbf{k}} \rho_{\mathbf{k}}(\tau) \right]$ with the following relation $\rho_{\mathbf{k}}(\tau) = \rho_{\mathbf{k}}^{(\text{off-diag})}(\tau) + \rho_{\mathbf{k}}^{(\text{diag})}(\tau)$. Note, the intensity is $I(\omega) \sim \omega^2 |\mathbf{J}(\omega)|^2$ with $\mathbf{J}(\omega) = \int d\tau \mathbf{J}(\tau) e^{i\omega\tau}$ from the Fourier transformation and similar manner for $\mathbf{J}_{\text{inter/intra}}(\omega)$. The spectral weight is then defined to be $I^{\text{nth}} = \int_{(n-0.5)\omega_{\text{exc}}}^{(n+0.5)\omega_{\text{exc}}} d\omega I(\omega)$ for the nth order harmonic intensity.

In fact, interband transition via multi-photon excitation are expected to exhibit series of multiple sharp resonance-like profiles at $2|\mu|/E_{ph} = n$ (n is an integer). In the perturbative regime without considering the scattering process, such profile has been theoretically predicted for $I_x^{5\omega}$ by the recent theoretical study [PRB 99, 195407(2019)], which we have drawn in Fig. R2.

However, intense laser excitation creates high density of photocarriers in graphene, which enables rapid electron scattering process. Then, series of sharp resonance-like profiles can be drastically broadened and merged together, forming one resonance-like profile located at $2|\mu|/E_{ph} = \text{non-integer}$. For example, even in the perturbation regime, the recent experimental and theoretical results [Fig. 4c in Nature Photonics **12**, 430–436 (2018) and New J. Phys. **16**, 053014 (2014)] shows that the 3rd harmonic intensity becomes the highest when $2|\mu|/E_{ph}$ is away from 2.

In order to investigate such possibility for $I_x^{5\omega}$ in our case, we have theoretically calculated 5th harmonic intensity as a function of chemical potential (Fig.R1) by employing quantum master equation which fully takes into account the scattering process. Fig. R1 shows how massless Dirac states around the Dirac point in the momentum space contributes to generate 5th harmonic current $J_x^{5\omega}$ in graphene under linearly polarized excitation along the x-direction (E_x). Dashed lines are constant energy contours on a Dirac cone, which describes electronic states vertically separated by energy mE_{ph} (m is an integer). Real and imaginary parts of $J_x^{5\omega}$ (Fig. R1b and R1c, respectively) are most strongly generated from

Dirac states in the region between $2E_{\text{ph}}$ and $4E_{\text{ph}}$. Depending on the photon number required for multi-photon transitions and the azimuthal angle, Dirac states generate $J_x^{5\omega}$ with characteristic sign and magnitude, which destructively interfere when radiating $I_x^{5\omega}$. As $2|\mu|$ increases, Pauli blocking sequentially disables contribution from Dirac states starting from near the Dirac point to higher energy. As shown in Fig. R1, $I_x^{5\omega}$ increases as the destructive interference is partly eliminated while $I_x^{5\omega}$ eventually disappears as all the resonant Dirac states are disabled. Our theoretical calculation (Fig. R1a) also shows that the maximum intensity is exhibited around at $2|\mu|/E_{\text{ph}} = 3.4$, which is the most common value for $2|\mu|/E_{\text{ph}}$ in our experimental data. We have included the discussion above in the revised manuscript.

Reviewers' Comments:

Reviewer #1:

Remarks to the Author:

In their revised manuscript, the authors carefully revisited the interpretations of their experimental findings. The authors did clarify many of my concerns. In my opinion, they have positively increased the scope of their work by clarifying the ultrafast dynamics of the massless Dirac fermions in gate tunable graphene. They also elucidated the interference between quantum pathways beyond the perturbative regime, in which both interband and intraband transition can be effectively coupled under elliptically polarized strong laser fields.

Their work will be highly interesting to the solid-HHG community and will stimulate series of works on the study of laser-field-driven dynamics with an extra gate control. As such, I recommend publication of the present manuscript in Nature Communications.

Reviewer #2:

Remarks to the Author:

The authors have revised their manuscript taking into account the comments of both reviewers. I believe that the revised manuscript have addressed all my previous comments, and I am happy to recommend it publication.

REVIEWER COMMENTS

1. Reply to Reviewer #1

Original comment (1):

General remarks of Reviewer 1:

In their revised manuscript, the authors carefully revisited the interpretations of their experimental findings. The authors did clarify many of my concerns. In my opinion, they have positively increased the scope of their work by clarifying the ultrafast dynamics of the massless Dirac fermions in gate tunable graphene. They also elucidated the interference between quantum pathways beyond the perturbative regime, in which both interband and intraband transition can be effectively coupled under elliptically polarized strong laser fields.

Their work will be highly interesting to the solid-HHG community and will stimulate series of works on the study of laser-field-driven dynamics with an extra gate control. As such, I recommend publication of the present manuscript in Nature Communications.

Our reply: We sincerely appreciate the referee for her/his appreciation of the improvements in our revised manuscript and for her/his recommendation on our work for the publication in Nature Communications. Following her/his valuable comments and advice, our manuscript was significantly improved.

2. Reply to Reviewer #2

Original comment (1):

The authors have revised their manuscript taking into account the comments of both reviewers. I believe that the revised manuscript have addressed all my previous comments, and I am happy to recommend it publication.

Our reply: We sincerely appreciate the referee for her/his appreciation of the improvements in our revised manuscript and for her/his recommendation on our work for the publication in Nature Communications. Following her/his valuable comments and advice, our manuscript was significantly improved.